# Multi-functional DNA nanostructures that puncture and remodel lipid membranes into hybrid materials

Oliver Birkholz [1], Jonathan R. Burns[2], Christian P. Richter[1], Olympia E. Psathaki[1], Stefan Howorka[2] & Jacob Piehler [1]

Synthetically replicating key biological processes requires the ability to puncture lipid bilayer membranes and to remodel their shape. Recently developed artificial DNA nanopores are one possible synthetic route due to their ease of fabrication. However, an unresolved fundamental question is how DNA nanopores bind to and dynamically interact with lipid bilayers. Here we use single-molecule fluorescence microscopy to establish that DNA nanopores carrying cholesterol anchors insert via a two-step mechanism into membranes. Nanopores are furthermore shown to locally cluster and remodel membranes into nanoscale protrusions. Most strikingly, the DNA pores can function as cytoskeletal components by stabilizing autonomously formed lipid nanotubes. The combination of membrane puncturing and remodeling activity can be attributed to the DNA pores' tunable transition between two orientations to either span or co-align with the lipid bilayer. This insight is expected to catalyze the development of future functional nanodevices relevant in synthetic biology and nanobiotechnology.

[1] Department of Biology and Center for Cellular Nanoanalytics (CellNanOs), University of Osnabrück, Barbarastr. 11, 49076 Osnabrück, Germany. [2] Department of Chemistry, Institute of Structural and Molecular Biology, University College London, 20 Gordon Street, London, WC1H OAJ, UK. Correspondence and requests for materials should be addressed to S.H. (email: s.howorka@ucl.ac.uk) or to J.P. (email: piehler@uos.de)

Lipid bilayer-enclosed compartments of defined permeability, size, and shape are essential in biology. They have been key in the evolution of prokaryotic cells[1] and are the hallmark of eukaryotic cells containing a network of interconnected organelles. The functionality of such membrane systems relies on an array of dedicated membrane proteins, which make up more than 1/3 of the proteome of a eukaryotic cell[2]. Integral, i.e. membrane-spanning, proteins play a crucial role in fundamental cellular processes such as membrane transport, cellular communication, and energy conversion. These processes are spatiotemporally controlled by the lateral organization and local shape of biological membranes by dedicated peripheral membrane proteins in conjunction with cytoskeletal proteins[3, 4]. However, an important contribution of integral membrane proteins to such membrane remodeling is emerging[5, 6], often caused by protein oligomerization[7, 8], protein–lipid interactions[9] and/or lipid phase segregation[10]. Such phenomena have frequently been related to a geometric mismatch of protein and membrane hydrophobicities[6, 11].

Synthetic biology has considerable interest to rationally engineer the complex functions of membrane proteins for biotechnological applications. However, the endeavor is partly thwarted by the notorious difficulty of producing and handling membrane proteins as well as their intricate physicochemical properties[12]. A potential alternative are DNA nanostructures which are currently considered the simplest route towards rational nanoscale design[13–18]. Indeed, DNA-based transmembrane nanopores (NP) have recently mimicked integral channel proteins[19–26]. The synthetic NPs are composed of a bundle of interconnected DNA duplexes to enclose a central hollow channel, and additionally carry lipid anchors to achieve membrane insertion. Nanopores can hence perforate lipid bilayers to facilitate transport of water-soluble molecules and act as cytotoxic agents[21] or molecular valves for drug delivery[23].

The unique potential of DNA NP in synthetic biology and nanobiotechnology[27] has, however, not been exploited due to unresolved fundamental questions about their interaction with membranes. While insertion of NPs in a transmembrane geometry has been deduced from single-channel current recordings[23, 24] and fluorophore-release assays[23], the mechanism of how NP bind and puncture membranes is still unclear. Similarly, it is not known whether NPs, once inserted into membranes, arrange into higher-order assemblies, or alter the morphology of the surrounding lipid bilayer as observed for some integral membrane proteins[4, 28, 29].

Here, we use quantitative single-molecule localization microscopy (SMLM) to unravel membrane insertion and spatiotemporal dynamics of prototypical NPs in the context of lipid bilayers. We combine SMLM with polymer-supported membranes (PSMs) assembled on a high-density polyethylene glycol (PEG) polymer cushion that separates the membrane from the underlying glass surface[30, 31]. In PSMs, membrane-embedded macromolecules do not interact with the glass surface, which has been successfully exploited to analyze diffusion and interaction of reconstituted integral membrane proteins[30, 32, 33].

Our study is conducted with a DNA-based NP[23] that carries up to three cholesterol anchors (NP-3C)[23] and a single fluorophore for spectroscopic and microscopic detection. Our quantitative studies establish that (i) membrane insertion proceeds via a two-step mechanism, and that (ii) inserted DNA NPs laterally cluster driven by hydrophobic mismatch. Furthermore, we discover that (iii) DNA NP remodel excess vesicular structures on the PSM surfaces into membrane protrusions and, most strikingly, support the formation of ultrathin lipid tubes by anchoring to the membrane inside the tubes' lumen. The multifunctional role of NPs as bilayer-spanning channels and peripheral membrane-remodeling scaffolds does not have a biological equivalent and will pave the way for new applications of engineered DNA nanostructures in synthetic biology and biotechnology, e.g. for cell-like networks, drug-delivery or imaging vesicles.

## Results

**Design and formation of the DNA nanopores.** Our study employed an archetypical DNA NP (Fig. 1a) composed of six hexagonally arranged DNA duplexes that are interlinked via hairpins at their termini. The pore is 9 nm in height, 5 nm in outer width, and has a 2 nm wide-channel lumen. Every other DNA duplex carries a cholesterol anchor on the outside to yield

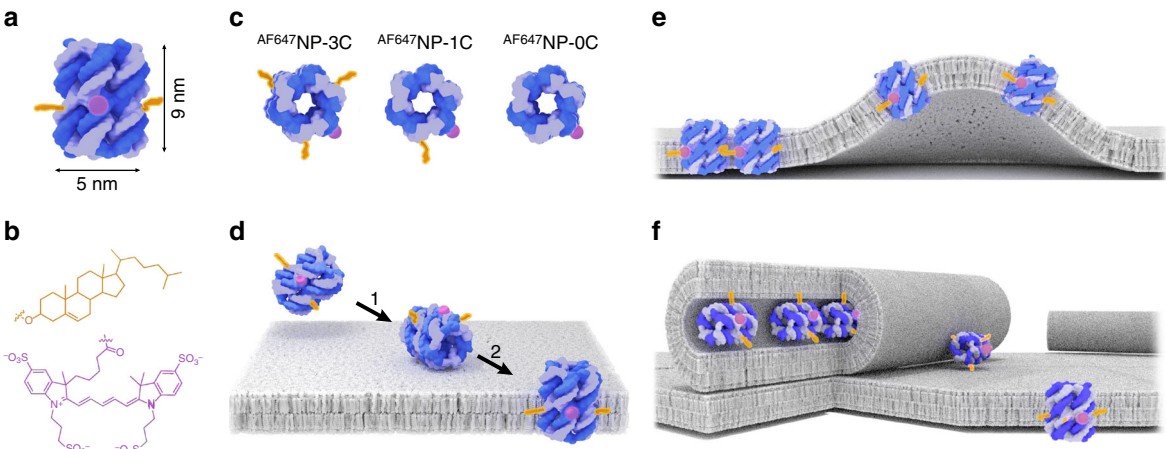

**Fig. 1** Structural and functional architecture of the DNA nanopores. **a** Side-view and external dimensions of NP-3C carrying three cholesterol anchors (orange) and the fluorophore Alexa Fluor 647 (AF647 - purple). **b** Chemical structure of cholesterol moieties (orange) for binding and insertion into membranes, and the Alexa Fluor 647 fluorophore (purple) used for detecting individual pores in the membranes. **c** Top-down view of NP-3C, NP-1C, and NP-0C with 3, 1, and no cholesterol anchors, respectively. **d–f** Mechanisms of DNA NP insertion, clustering and membrane-reshaping identified in this study. **d** Anticipated two-step integration of NP-3C into a lipid bilayer. **e** Formation of dynamic NP-3C clusters and enrichment in curved membrane structures. **f** Proposed model for the NP-3C-induced formation of lipid nanotubes with luminally attached pores in case of excess lipid on the membrane surface. For reasons of visual simplicity, **d-f** do not account for the reorganization of the lipids at the interface to the NPs. Lipids are likely re-oriented to position the polar head-group closer to those DNA segments that do not carry a cholesterol anchor[19]

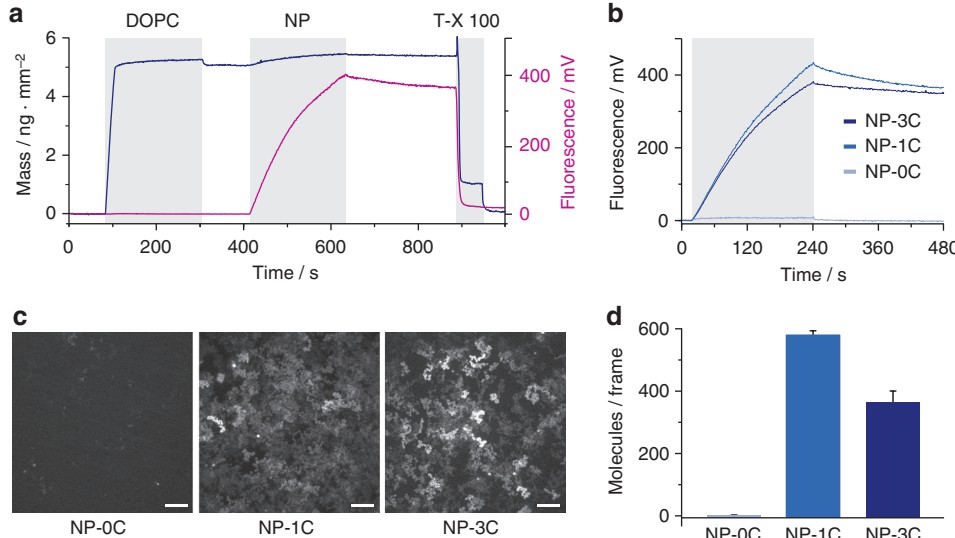

**Fig. 2** Cholesterol-dependent NP binding to PSMs. **a, b** Kinetics of DNA NP binding to polymer-supported membranes (PSM) as monitored by simultaneous total internal reflection fluorescence and reflectance interference spectroscopy (TIRFS-RIf) detection. **a** Spontaneous PSM formation upon binding of 1,2-dioleoyl-sn-glycero-3-phosphocholine (DOPC) vesicles was confirmed by label-free RIf detection (dark blue line). The kinetics of 100 nm $^{AF647}$NP-3C binding to the PSM could be resolved with high signal-to-noise by TIRFS (magenta line), while the surface coverage could be estimated from the RIf signal. Subsequently, the surface was regenerated by a detergent wash to allow repeated experiments under the same conditions. **b** Comparison of the TIRFS signal obtained for 100 nm $^{AF647}$NP-0C, $^{AF647}$NP-1C, and $^{AF647}$NP-3C. **c, d** DNA NP binding quantified by TIRFM. **c** Representative maximum intensity projection of 100 frames acquired after incubating PSMs with 5 pm NP-0C, NP-1C, and NP-3C, respectively. Scale bars: 5 μm. **d** Quantification of detected single-molecules in a full frame (512×512 pixel) after incubation of the PSM with 5 pm of NPs for 10 min (mean and s.d. of $n = 3$ experiments)

pore NP-3C (Fig. 1a, b)[23]. NP variants with a single cholesterol moiety (NP-1C) or no lipid anchor (NP-0C) were used as controls (Fig. 1c). For detection and reliable quantification, a single Alexa Fluor 647 fluorophore (Fig. 1b) was attached to one of the unmodified strands of the NPs yielding $^{AF647}$NP-3C, $^{AF647}$NP-1C, and $^{AF647}$NP-0C (Supplementary Fig. 1 and Supplementary Tables 1 and 2 for DNA sequences). The pores were assembled from a mixture of synthetic DNA oligonucleotides via a single annealing step to produce a single product as shown by gel electrophoresis (Supplementary Fig. 1). The fluorophores did not interfere with pore insertion into lipid bilayers, as shown by studies with small unilamellar vesicles (Supplementary Table 3).

**Fast membrane tethering of NPs via single cholesterol anchor.** To elucidate the kinetics of membrane binding of NP, we employed real-time monitoring by simultaneous total internal reflection fluorescence spectroscopy (TIRFS) and label-free detection by reflectance interference spectroscopy (RIf) in a flow-through system. Prior to examining NP-binding kinetics, RIf analysis established the successful assembly of PSMs composed of 1,2-dioleoyl-sn-glycero-3-phosphocholine (DOPC, Fig. 2a). The RIf and TIRFS signals of $^{AF647}$NP-3C that was subsequently added suggested a rapid membrane-association of the DNA NP (Fig. 2a and Supplementary Fig. 2) that required cholesterol-modified NP because $^{AF647}$NP-0C without a lipid anchor did not bind. From the TIRFS binding curves of $^{AF647}$NP-3C, the rate constants for association ($k_a$) and dissociation ($k_d$) were determined. The $k_a$ of ~1.5·10[5] $M^{-1}$ $s^{-1}$ was in good agreement with the $k_a$ observed for lipid-anchored proteins[34], while the $k_d$ of 4.1·10$^{-4}$ $s^{-1}$ indicated quasi-irreversible anchoring into the membrane. The resulting equilibrium dissociation constant $K_D$ of ~5 nm (Supplementary Table 4) implied tight interaction of DNA NP with the membrane. The kinetics were dependent on the amount of DNA NP, as concentrations above 100 nm decreased $k_a$ by 1/3

(Supplementary Table 4), likely due to electrostatic repulsion between the negatively charged NPs. Indeed, for a concentration of 300 nm, the density of membrane-anchored NPs reached ~5000 NPs μm$^{-2}$, equivalent to one NP per 14×14 nm$^2$ as calculated from the RIf signal of ~0.8 ng mm$^{-2}$. However, application of such high DNA NP concentrations were only required for kinetic analysis of membrane anchoring, while more than 1000-fold lower densities were used for studying the spatiotemporal organization within the membrane by single-molecule fluorescence microscopy. For $^{AF647}$NP-1C with one cholesterol, very similar membrane association kinetics was observed (Fig. 2b, Supplementary Table 4). This suggests that solely a single anchor is required for initial membrane docking to result in a horizontal, non-membrane spanning orientation (Fig. 1d). The dissociation kinetics of $^{AF647}$NP-1C was - however - enhanced compared to $^{AF647}$NP-3C, likely due to less stable anchoring by the single cholesterol.

Efficient anchoring of both $^{AF647}$NP-3C and $^{AF647}$NP-1C was further confirmed by total internal reflection fluorescence microscopy (TIRFM). Compared to the kinetic experiments, a much lower DNA NP concentration of 5 pm was applied to visualize cholesterol-mediated anchoring at the single-molecule level. Indeed, time lapse imaging showed lateral diffusion of individual $^{AF647}$NP-3C and $^{AF647}$NP-1C pores, while cholesterol-free $^{AF647}$NP-0C did not bind (Fig. 2c and Supplementary Movie 1). The extent of DNA NP membrane-binding was quantified by localizing individual diffraction-limited signals in each frame of a time-lapse acquisition. This yielded a similar density of such "localized molecules" for $^{AF647}$NP-3C and $^{AF647}$NP-1C molecules, respectively (Fig. 2d). The slightly reduced density observed for $^{AF647}$NP-3C compared to $^{AF647}$NP-1C can be attributed to the method for identifying localized molecules, and the tendency of NP-3C to form clusters (see chapter "DNA NPs form dynamic nanoclusters"). Thus, clusters comprising multiple NPs are detected as single localized molecules, effectively reducing the inferred density.

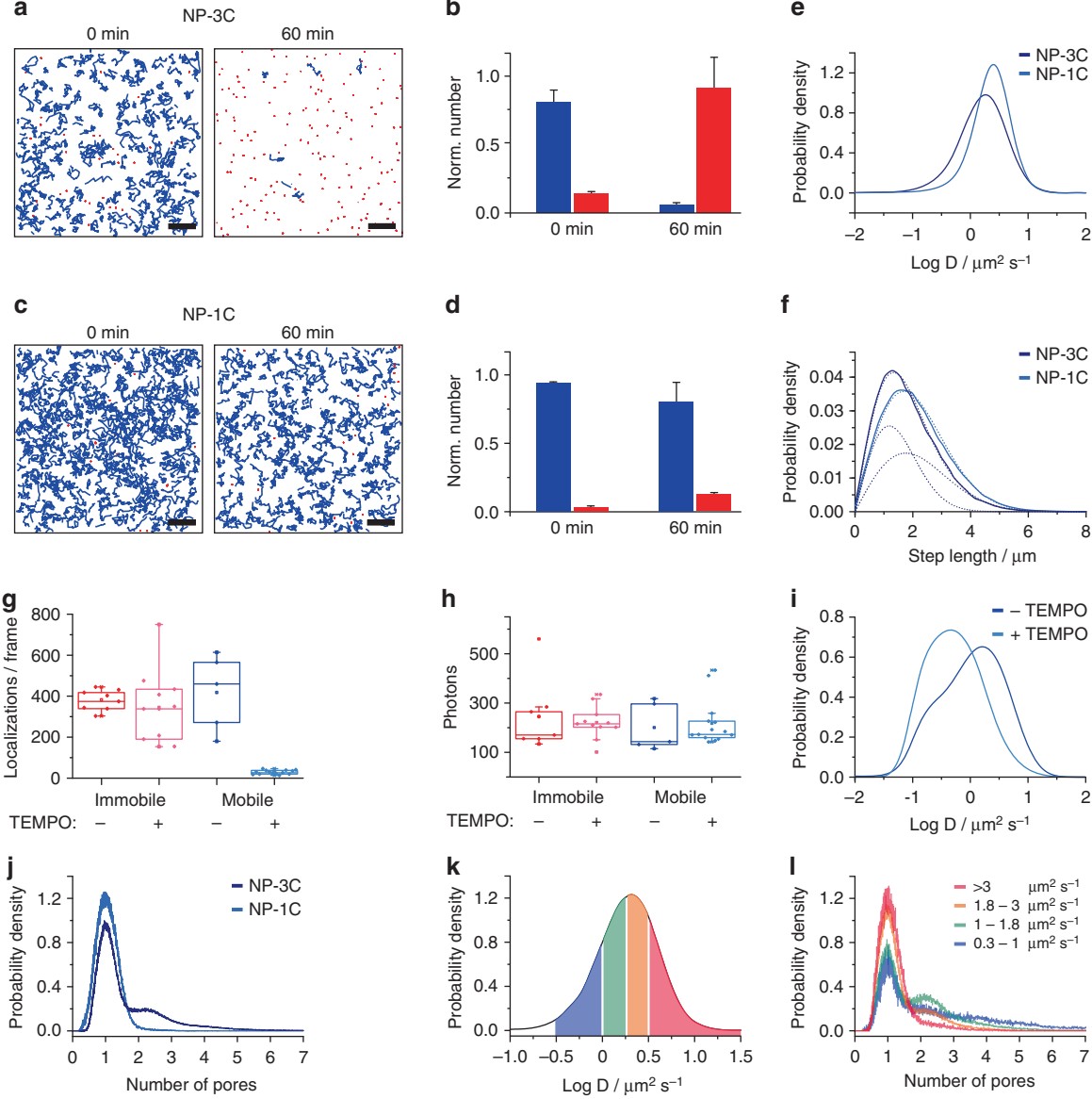

**Fig. 3** Single-molecule analysis of NP-1C and NP-3C mobility in PSM. **a–d** Time-dependent changes in nanopore mobility quantified by single-molecule analysis of NP-3C (**a**, **b**) compared to NP-1C (**c**, **d**). **a**, **c** Trajectory maps of mobile (blue) and immobile (red) [AF647]NP-3C (**a**) and [AF647]NP-1C (**c**) molecules at time points of 0 min (left) and after 60 min (right). Scale bars: 5 μm. **b**, **d** Quantification of the relative fraction of mobile and immobile NPs for [AF647]NP-3C (**b**) and [AF647]NP-1C (**d**) between both time points (mean and s.d. of n = 3–5 experiments). **e** Probability density function (pdf) for the diffusion coefficient of [AF647]NP-1C and [AF647]NP-3C trajectories at 0 min (n = 6800 and 3900 trajectories over ten frames long, respectively). **f** Step length distribution diagram (30 frames, 19 ms) of [AF647]NP-1C and [AF647]NP-3C and fit by a two-component Rayleigh model (dashed lines) of the same data set. **g** Number of mobile and immobile fluorescin-labeled NP-3C ([FAM]NP-3C) detected in each frame before and after addition of (2,2,6,6-tetramethylpiperidin-1-yl)oxyl (TEMPO). **h** Integrated intensity of detected molecules before and after addition of TEMPO. Each datapoint in **g**, **h** corresponds to a separate experiment (n ≥ 3). **i** Diffusion coefficients of mobile [FAM]NP-3C after addition of TEMPO (light blue) compared to the initial condition (dark blue). The distributions are based on n = 2000 and 3900 trajectories with a minimum of five frames, respectively. **j** Intensity distribution determined from [AF647]NP-1C (light blue) and [AF647]NP-3C (dark blue) trajectories normalized to single fluorophore intensities. **k** Distribution of the NP-3C diffusion coefficients. Individual photon traces of the color-coded regimes were extracted and separately analyzed by their fluorescence intensity like described for **j**. **l** Resulting probability density functions of the extracted diffusion regimes described in **k** in the same color-coding. The analysis in **j** and its subsets in **k**, **l** rely on 8800 and 19,700 trajectories (>10 steps) of [AF647]NP-1C and [AF647]NP-3C, respectively

## Membrane perforation by DNA NP is rate-limiting.

Single-molecule imaging of [AF647]NP-3C revealed time-dependent changes in its diffusion behavior, suggesting a re-orientation into a membrane-spanning state (Fig. 1d). In the first minutes after addition of [AF647]NP-3C, individual NPs were largely laterally mobile as implied by the motion blur, even though some brighter spots suggested immobile NPs (Fig. 2c and Supplementary Movie 2). Quantification by single-molecule tracking in

combination with a spatiotemporal identification algorithm[35] established that more than 80% of [AF647]NP-3C exhibited free Brownian motion as judged from a linear mean square displacement (MSD) diagram (Supplementary Fig. 3). However, 1 h after NP-3C binding to the membrane, the mobility drastically decreased to more than 80% stationary signals (Fig. 3a, b and Supplementary Movie 2). This fraction likely represents immobile membrane-spanning pores that interact with the polymer layer

underneath the lipid bilayer. This DNA NP state is further analyzed by SMLM (see next section). By contrast, only minor changes in mobility were observed for $^{AF647}$NP-1C (Fig. 3c, d). The molecules remained largely mobile even 2 h after binding to the membrane (Supplementary Fig. 4 and Supplementary Movie 3), corroborating that immobile $^{AF647}$NP-3C DNA NPs reflect membrane-spanning DNA NPs.

A more nuanced picture was obtained when comparing the diffusion coefficient $D$ for the mobile fraction of $^{AF647}$NP-1C and $^{AF647}$NP-3C, which was derived from single-particle trajectories using MSD and step length histogram analyses. While the value of $D$ for $^{AF647}$NP-1C at 2.5 $\mu m^2 s^{-1}$ (Fig. 3e, Supplementary Fig. 3) was in the range of lipid diffusion within PSMs[30], only $^{AF647}$NP-3C yielded two components with a fast diffusion $D = 2.5$ $\mu m^2 s^{-1}$ and slow diffusion at $D = 1.0$ $\mu m^2 s^{-1}$ (Fig. 3f). The slow fraction therefore likely represents membrane-spanning DNA pores, as the diffusion was close to single-transmembrane α-helical proteins or β-barrel pores in PSMs[30, 33]. In support of this interpretation, control experiments with bilayers directly fused onto glass surfaces revealed dramatically reduced insertion rate for $^{AF647}$NP-3C and an almost complete absence of the slow diffusing pore fraction (Supplementary Fig. 5).

These results strongly indicate that NP-1C and NP-3C initially docks to the membrane via a single cholesterol anchor (Fig. 1d, step 1) as implied by the fast diffusing fraction. In a second step, NP-3C re-orients and inserts into the membrane (Fig. 1d, step 2) to exhibit slower diffusion. The transition was not possible for NP-1C as it lacked multiple cholesterol anchors required for complete membrane insertion. To confirm that slower diffusion of immobile fraction NP-3C was due to a membrane-spanning state, we probed the differential accessibility of an NP-tethered fluorescence dye with quenching. The fluorescein moiety was strategically positioned within the hydrophobic belt of NP-3C ($^{FAM}$NP-3C) such that only membrane insertion would protect the dye from a water-soluble quenching agent (2,2,6,6-tetramethylpiperidin-1-yl)oxyl (TEMPO)[36]. In support of a membrane-spanning pore, the number of immobile $^{FAM}$NP-3C remained constant in the presence of TEMPO, while the mobile fraction was strongly reduced (Fig. 3g). The fluorescence intensity of the unquenched NPs remained unaltered (Fig. 3h). In further agreement, TEMPO led to selective removal of fast mobile, and assumed membrane-docked $^{FAM}$NP-3C, while the slower diffusing and membrane-spanning pores were preserved (Fig. 3i).

**DNA NPs form dynamic nanoclusters**. The observed fluorescence intensities for both mobile and immobile $^{AF647}$NP-3C signals were high and relatively broadly distributed (Fig. 2c). To investigate whether this represents an oligomeric state, the mobile trajectories of the NPs were subjected to an intensity-based analysis. A significant oligomer fraction was found for $^{AF647}$NP-3C but not $^{AF647}$NP-1C (Fig. 3j), implying that clustering occurred from membrane-spanning pores. In further support, the oligomeric state of $^{AF647}$NP-3C strongly correlated with the diffusion coefficient, as higher oligomers showed slower diffusion (Fig. 3k, l).

In complementary analysis, the oligomeric state of the immobile fraction was investigated by single-molecule photobleaching (Fig. 4a, Supplementary Movie 4). Photobleaching steps and fluorescence intensity levels were identified by an automated step transition and state identification (STaSI) algorithm[37], thereby yielding the number of individual DNA NPs within each nanocluster (Fig. 4b). The robustness of the STaSI algorithm was corroborated by a linear correlation between the number of photobleaching steps and the total intensity of the spots (Fig. 4c). These analyses clearly confirmed oligomerization of $^{AF647}$NP-3C

after insertion into PSM with more than 30% of immobile spots containing at least two DNA NPs (Fig. 4d), which is likely an underestimate due to the effective degree of labeling by photobleaching and bleaching events not detected by the algorithm. In further support of oligomerization, an elevated density of $^{AF647}$NP-3C within the PSM yielded larger sizes for the immobile clusters (Fig. 4d). Pores are likely immobilized by the increased friction caused by the interaction with the fatty acid moieties that anchor the PSM to the polymer cushion. Clustering at the very low DNA NP density used in these experiments could alternatively be explained by the considerable geometric mismatch between the small-sized cholesterol groups of DNA NPs and the much wider hydrophobic core within the lipid bilayer ("hydrophobic mismatch"). Under these conditions, cholesterol-mediated hydrophobic contacts between DNA NPs are preferred over interactions with lipids leading to liquid phase separation between lipids and the membrane-spanning nanostructures[38].

Clustering driven by hydrophobic mismatch and liquid phase separation implicates that the cluster composition is dynamic and involves constant exchange with the mobile NP fraction. To validate this concept, we explored the exchange dynamics of clusters by fluorescence recovery after photobleaching (FRAP). For this purpose, $^{AF647}$NP-3C pores within a region of interest (ROI) were completely photobleached and the same ROI was imaged again 15 min afterwards. Strikingly, immobile $^{AF647}$NP-3C nanoclusters were present at the same density before and after photobleaching (Fig. 4e), several of which were either overlaying or in very close proximity. This result confirmed that membrane-inserted DNA NP do not aggregate irreversibly, but rather reversibly assemble into dynamic oligomers. The spatial correlation of nanocluster positions before and after photobleaching was quantified by particle image cross-correlation spectroscopy (PICCS)[39]. A correlated fraction of $9.2 \pm 0.6\%$ with an average correlation length of $22.5 \pm 2.2$ nm (mean values + s.e.m.) was obtained (Fig. 4f). As this distance is close to the localization precision, one can infer that the fraction of nanoclusters did not change their position within the time frame of the experiment. The remaining NP nanoclusters disassembled and re-assembled at other positions. These observations are in line with the interpretation that NP-3C clustering is driven by hydrophobic mismatch and transient liquid phase separation. Similar behavior has been previously observed for lipid phase separation in tethered PSM[40].

**DNA NPs prefer curved membranes and remodel bilayer shape**. Hydrophobic mismatch is a typical feature of integral membrane proteins which has been implicated in oligomerization, but also remodeling of the membrane shape[5, 6, 41]. As oligomerization was found for DNA NP, we tested whether the DNA structures also reshape the bilayer. We first probed the interaction of $^{AF647}$NP-3C with existing membrane protrusions that were formed by excess lipid on the surface of PSM under less stringent washing conditions (Supplementary Movie 5). Fluorescence imaging at low concentrations (5 pM) of $^{AF647}$NP-3C revealed strong binding to the curved membrane protrusions (Supplementary Movie 5). To explore whether DNA NPs also actively remodel the bilayer shape, membranes doped with lipid marker Oregon Green 488 1,2-dihexadecanoyl-sn-glycero-3-phosphoethanolamine ($^{OG488}$DHPE) were briefly incubated with an elevated concentration of $^{AF647}$NP-3C (5 nM), followed by washing out unbound DNA NPs. At this 1000-fold increased concentration, membrane coverage with $^{AF647}$NP-3C appears rather homogeneous since individual nanoclusters cannot be properly resolved (Fig. 5a). Time-lapse confocal imaging, however, revealed that in the presence of $^{AF647}$NP-3C, membranes

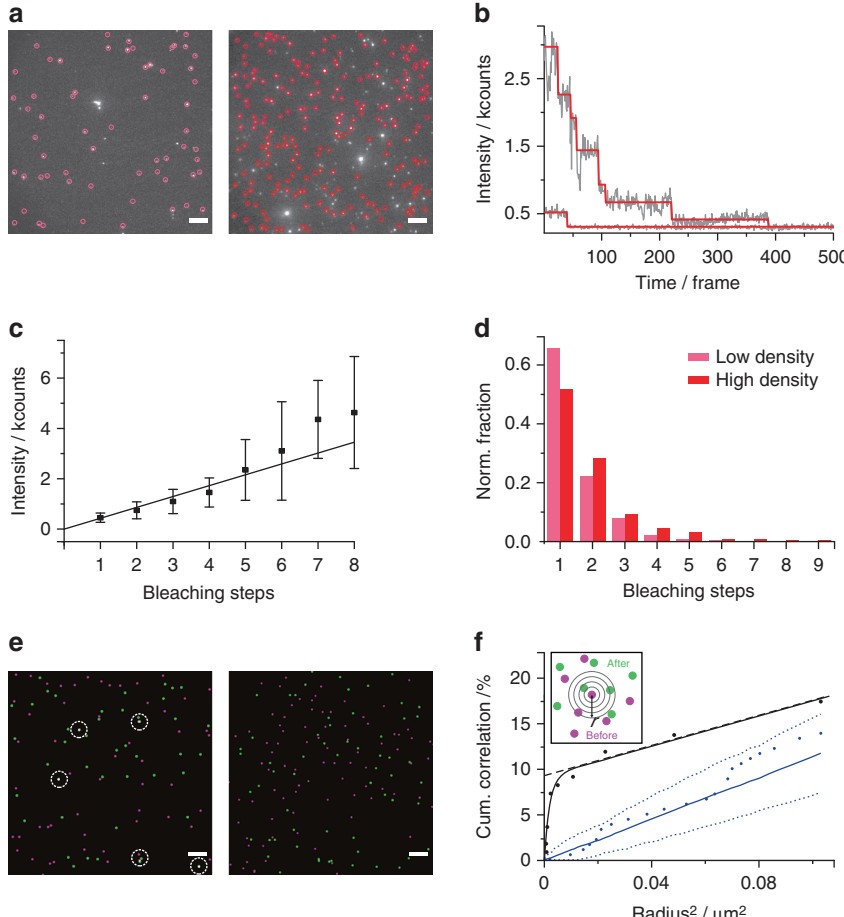

**Fig. 4** Dynamic clustering of $^{AF647}$NP3-C explored by photobleaching analysis. **a** Photobleaching analysis at low (left: 0.47 μm$^{-2}$) and high (right: 1.94 μm$^{-2}$) densities of immobile $^{AF647}$NP-3C inserted into DOPC PSM. Signals that were detected and used for the analysis are encircled. Scale bars: 5 μm. **b** Typical intensity traces of individual signals showing single or multi-step intensity decays by photobleaching with an overlay of the STaSI trace shown in red. **c** Correlation of the intensity of localized signals in the first frame against the number of detected bleaching steps by the STaSI algorithm ($n = 10$, mean and s.d.). **d** Histogram of bleaching steps for low and high densities. The data in **c** and **d** relies on analyzing $n = 5822$ clusters at high density and 1397 clusters at low density. **e**, **f** Spatial correlation of $^{AF647}$NP-3C clusters explored by fluorescence recovery after photobleaching (FRAP) and particle image cross-correlation spectroscopy (PICCS). **e** Left: Localized immobile $^{AF647}$NP-3C clusters before (magenta) and 15 min after (green) complete photobleaching of the shown area. $^{AF647}$NP-3C clusters co-localizing in both images are highlighted by white circles. Right: negative control showing the overlay of immobile $^{AF647}$NP-3C signals from two separate, uncorrelated areas. Scale bars: 5 μm. **f** PICCS analysis of clusters localized before and after photobleaching (black) and the negative control (blue solid line) including the boundaries of a 95% confidence interval (blue dotted lines). The intercept of the linear contribution to the curve (black dashed line) corresponds to the correlated fraction α. The concept of PICCS is graphically depicted in the inset

protrusions were successively increasing in size and number (Fig. 5a and Supplementary Movie 6). This suggests that insertion and subsequent clustering of NP-3C leads to membrane remodeling as, e.g., observed for proteins containing BAR domains[42]. By comparing the relative densities of $^{AF647}$NP-3C and the membrane marker $^{OG488}$DHPE, we observed a twofold enrichment of NP-3C in curved compared to planar membrane regions (Fig. 5b, c). Rapid exchange of $^{AF647}$NP-3C between protrusions and planar membranes as confirmed by FRAP experiments (Supplementary Fig. 6) corroborated reversible segregation and clustering of DNA NPs in membranes during remodeling. These experiments imply that DNA NPs promote formation of membrane protrusions due to preferential anchoring, integration or clustering in curved membranes as depicted in Fig. 1e.

**NP-3C induces formation of lipid nanotubes**. As another striking observation, $^{AF647}$NP-3C at low concentration induced formation of characteristic, long tubular membrane structures.

The structures were apparent from a distinctive one-dimensional diffusion of individual $^{AF647}$NP-3C pores (Fig. 6a left, Supplementary Movie 7). Nanotubes formation could be enhanced by the presence of excess lipid material, for example caused by addition of NP-3C prior to vesicle fusion. This yielded different tube geometries including long, isolated tubes spanning several tenth of microns as well as strongly kinked and branched network-like structures (Supplementary Fig. 7). While DNA NPs showed rapid one-dimensional diffusion along isolated lipid nanotubes, immobile DNA NPs were observed at the edges and branches, suggesting different membrane integration states. The majority of the lipid nanotubes were physically connected via sparse contact points to the planar membrane, as demonstrated in FRAP experiments with $^{OG488}$DHPE by the rapid fluorescence recovery of the nanotubes from lipid exchange with the underlying membrane (Supplementary Fig. 8 and Supplementary Movie 8). While $^{OG488}$DHPE in the planar membrane could freely diffuse beneath the lipid nanotubes (Supplementary Fig. 9), pores regularly bounced off the tubes (Supplementary Movie 9),

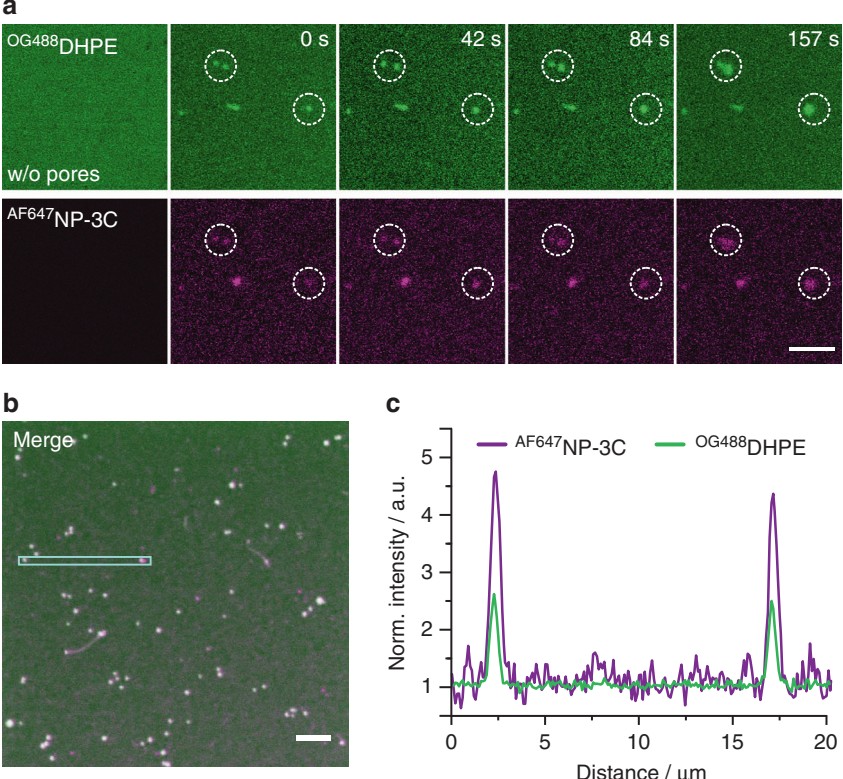

**Fig. 5** Membrane remodeling by NP-3C. **a** Time-lapse confocal fluorescence imaging of $^{AF647}$NP-3C (magenta) bound to membrane protrusions from a PSM that was doped with the lipid marker Oregon Green 488 1,2-dihexadecanoyl-sn-glycero-3-phosphoethanolamine ($^{OG488}$DHPE (- green). The first column shows a representative area before the addition of $^{AF647}$NP-3C. Scale bar: 5 μm. **b** Overlay image of $^{AF647}$NP-3C (magenta) bound to the PSM stained with $^{OG488}$DHPE (green). Scale bar: 5 μm. **c** Fluorescence intensity profile across two representative membrane protrusions highlighted in panel **b**. Both fluorescence channels were normalized to the basal fluorescence levels within the planar PSM

suggesting a close proximity of nanotube and membrane. The spatiotemporal organization of DNA NPs in these tubular structures was resolved by tracking and localization microscopy (TALM)[43]. To this end, the positions of individual $^{AF647}$NP-3C molecules within 900 consecutive frames were superimposed into a single reconstructed superresolution image (Fig. 6a, right). Single-molecule tracking performed on this data set yielded trajectories that were strictly co-aligned within the superresolution image, thus establishing that the DNA NPs were stably anchored to the tubes (Fig. 6b). From the one-dimensional diffusion of NP-3C along nanotubes (Supplementary Fig. 10), a diffusion constant of 1 μm² s⁻¹ was obtained which is comparable to diffusion of the inserted pores. However, the diffusion constant could also be accounted for by NP that are inside the lipid nanotube cavity and tethered via the three cholesterol anchors to the surrounding cylindrical membrane (Fig. 1f).

To test whether pores can be inserted into the tube membrane, we added sulforhodamine B (SRhoB), which is known to translocate through the pores[23], and examined whether the fluorophore can diffuse via inserted pores inside the cavity. Indeed, SRhoB accumulated within NP-3C-induced lipid tubes and could be localized at the single-molecule level with nanometer precision (Supplementary Movie 10), implying that at least some pores were membrane-spanning. However, further data indicated the NP are also inside the cavity. For example, the area accessed by $^{AF647}$NP-3C and SRhoB was identical as analyzed by dual-color superresolution TALM images from 325 consecutive frames (Fig. 6c and Supplementary Movie 10). In particular, the overlay of single-molecule coordinates perpendicular to the tube axis revealed undistinguishable Gaussian

distributions for SRhoB and $^{AF647}$NP-3C (full-width at half maximum: 38 ± 3 nm for $^{AF647}$NP-3C and 45 ± 2 nm for SRhoB, Fig. 6d). Taking into account the localization precision (19 nm for AF647 and 25 nm for SRhoB, Supplementary Fig. 11), a tube diameter of <20 nm could be estimated from both markers. The formation of such ultrathin lipid nanotubes was directly confirmed by negative stain transmission electron microscopy (TEM). After adding NP-3C to lipid vesicles deposited on carbon films, networks of interconnecting lipid nanotubes with a diameter of 15–20 nm were observed (Fig. 6f, g) that were absent in control experiments without DNA NPs (Fig. 6h). Taken together, these results suggest that mobile NP-3Cs are located inside the lipid nanotubes with their cholesterol moieties axially inserted into the bilayer (Fig. 6e) allowing one-dimensional diffusion inside the lumen of the lipid nanotube. In contrast, immobile DNA NPs found at the kinks and branches of lipid nanotubes are probably inserted into the membrane, which is in line with the strong membrane bending properties of inserted DNA NPs and the uptake of SRhoB.

**Lipid nanotubes are stabilized by DNA NPs inside the lumen.** To further test whether NP-3Cs were located inside the lumen of lipid nanotubes, we probed the accessibility of the FAM fluorophore of $^{FAM}$NP-3C by addition of TEMPO. As fluorescence of the one-dimensional diffusion was not quenched (Supplementary Fig. 12), DNA NPs were likely protected due to their intratubular location. Yet, a small fraction of membrane spanning DNA NPs could not be ruled out by the assay, which could explain transport of SRhoB into the lumen of the lipid nanotube.

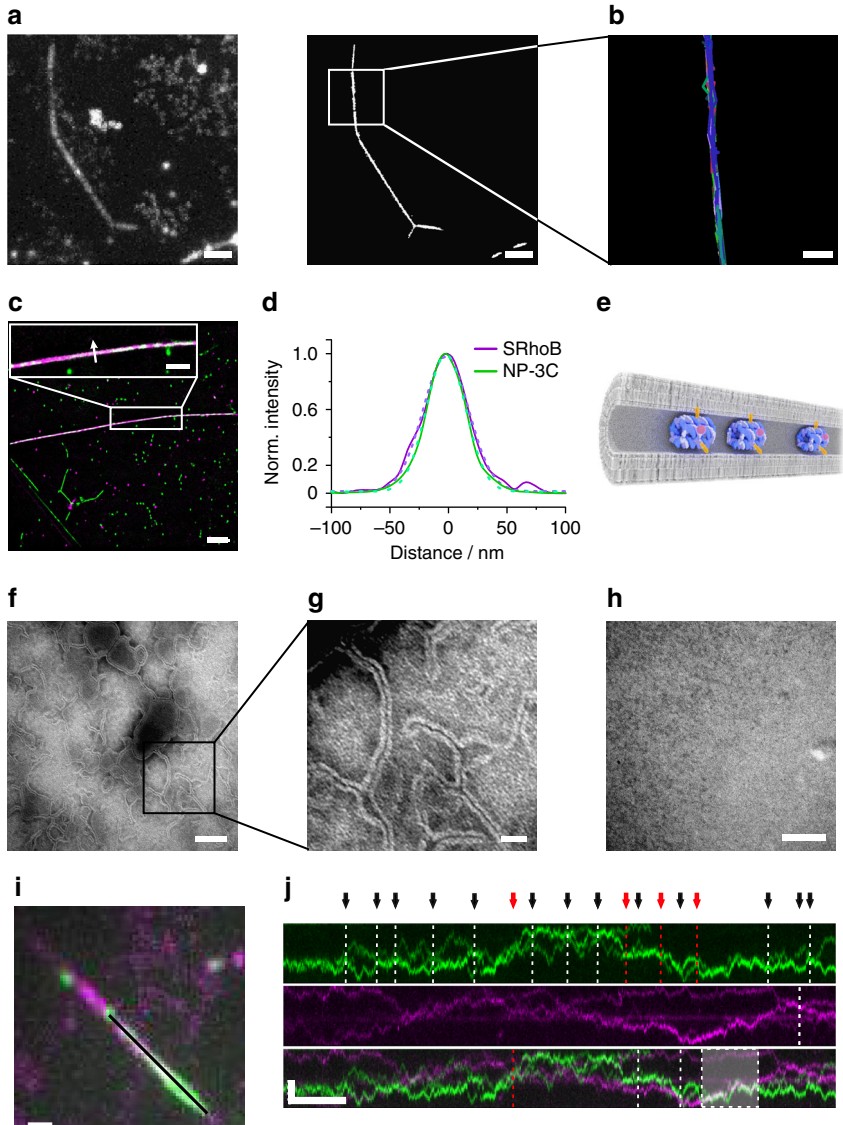

**Fig. 6** Formation of lipid nanotubes observed in presence of NP-3C. **a**, **b** Typical lipid nanotube on the surface of a PSM observed in presence of [AF647]NP-3C resolved by tracking and localization microscopy (TALM). **a** Maximum intensity projection (left) and superresolution TALM image (right) of [AF647]NP-3C diffusion in a lipid nanotube. Scale bars: 2 μm. **b** Trajectories of [AF647]NP-3C. Scale bar: 500 nm. **c** Dual-color superresolution TALM images of sulforhodamine B (SRhoB, magenta) and [AF647]NP-3C (green) obtained by rendering single-molecule localizations from 325 consecutive frames. Scale bar: 5 μm. **d** Profiles of localized AF647 and SRhoB signals across the tube depicted by the arrow with a Gaussian fit (dotted line). The full-width at half maximum (FWHM) for [AF647]NP-3C and SRhoB was $38 \pm 3$ nm and $45 \pm 2$ nm, respectively ($n = 6$ for each channel, mean and s.d). **e** Proposed architecture of NP-3C inside lipid nanotubes. **f**, **g** Representative negative stain transmission electron microscopy (TEM) revealing a network of ultrathin lipid tubes. Scale bar: 100 nm (**f**) and 25 nm (**g**). **h** Negative control TEM image of a membrane on a carbon film in absence of DNA NPs. Scale bar: 100 nm. **i** Maximum intensity projection of 150 consecutive frames of [AF647]NP-3C (green) and [Cy3]NP-3C (magenta) including a merged image. Scale bar: 1 μm. **j** Kymographs of [AF647]NP-3C (green) and [Cy3]NP3C (magenta) diffusion along the black line indicated in **i**, shown in separate and merged channels. Black arrows indicate collisions between individual NPs, red arrows highlight crossing events, while the shaded area shows elongated co-diffusion along the nanotubes. Scale bars: 2 μm (vertical), 500 ms (horizontal)

DNA NPs that are anchored via cholesterol inside the lipid nanotubes would be remarkable because they would support the structure of the lipid nanotubes, and thereby function as artificial cytoskeleton. If the model of DNA NPs, as depicted in Fig. 6e, was indeed correct, then head-on collisions between the internally confined pores would be more frequent than if the DNA NPs were perforating the membrane or being tethered to its outer side. In general, the probability of detecting random collisions between particles that change their direction by 180° is extremely low due to the sparse DNA NP densities used for single-molecule imaging

experiments, as well as the short observation times limited by photobleaching. However, the linear diffusion of DNA NPs within the highly confined lipid nanotubes would enhance the chance of detecting collisions. To unambiguously visualize such single-molecule collisions, we performed dual-color experiments with [Cy3]NP-3C and [AF647]NP-3C at low densities to spectrally discriminate individual DNA NPs diffusing along nanotubes (Fig. 6i and Supplementary Movie 10[1]). In strong support of intratubular localization, encounters of individual DNA NPs often led to bouncing off each other (Fig. 6j, upper panel) and

even co-diffusion for a prolonged time (Fig. 6j, lower panel and Supplementary Movie 11). Only in rare cases, crossing of individual pores along tubes was observed, which could be explained by DNA NPs anchored at the outside of the tube.

## Discussion

DNA NPs have recently been developed to mimic membrane proteins and open up new strategies to engineer transmembrane functions at the molecular level. Our study substantially advances the potential of DNA NPs by revealing multiple ways in which the DNA nanostructures interact with lipid bilayers, and by describing the physicochemical factors governing the interaction. As first highlight, we established a two-step insertion mechanism, where tethering of an NP to the membrane via one cholesterol anchor is followed by insertion into a transbilayer geometry requiring multiple anchors. Owing their significantly different diffusion coefficients, DNA NPs in membrane-tethered and transmembrane conformations could be clearly discriminated at the single-molecule level. While tethering into the membrane was a comparably fast process, the long lifetime of the membrane-anchored conformation suggests that membrane insertion of three-anchored pores overcomes a substantial energy barrier. Future engineering of DNA pores can exploit the insight by, for example, tuning the chemical nature or position of lipid anchors for faster membrane poration.

Our study found, as a second highlight, that triple cholesterol-anchored DNA NPs spatially segregate and cluster within the membrane, which was most likely the result of the hydrophobic mismatch with the lipid bilayer. However, these clusters are not formed irreversibly, but dynamically exchange DNA NPs with the surrounding lipid membrane. Moreover, the DNA NPs enriched at curved, compared to planar lipid bilayers, and promoted the formation of membrane protrusions. Similar features are highly relevant for biological functions of membrane proteins. For example, signaling across membranes frequently involves dynamic, spatiotemporally regulated clustering of receptors,[44, 45] which is intricately linked to endocytosis,[46, 47] a process, which fundamentally requires remodeling of membranes to increase curvature[48]. Such processes have recently been ascribed to a mismatch between protein hydrophobic surface and lipid bilayer thickness, as studied in detail for G-protein-coupled receptors[6].

As third highlight, our study discovered the ability of the triple-anchored NP to promote membrane curvature and, most strikingly, induce the formation of ultrathin lipid tube + DNA NP hybrid structures. These surprising functions required that excess lipid in vesicular structures was available at the PSM. We propose that membrane remodeling is mostly driven by energetically favorable anchoring of all three cholesterol anchors of NP-3C into membranes in a non-spanning geometry. Indeed, our finding of spatially confined diffusion strongly suggests that the DNA pores are inside the lumen of the elongated lipid structures (cf. Figure 6e) and are stabilized by the radially positioning its cholesterol anchors into the surrounding membrane, which allows rapid movement along the lumen of the lipid nanotube. Up until now, lipid nanotube formation has been achieved either via mechanical forces[49, 50], with nanosized scaffolds[51] or by proteins that polymerize into membrane-bending structures[52, 53] or induce membrane bending by local crowing[54]. Similarly, lipid membranes have also been shaped by membrane-floating DNA origami[55, 56]. Yet, formation of lipid nanotubes based on DNA scaffolds has not been reported. In contrast to previously described rigid scaffolds, DNA NPs seem to stabilize lipid nanotubes at very low density and in a highly dynamic manner as rapid diffusion along the lumen is maintained. Most strikingly, the DNA NPs are for the first time shown to act both an integral membrane pore as well as a cytoskeletal membrane-remodeling component. This unique dual function of DNA NPs is probably linked to their variable interaction with the membrane allowing peripheral and transmembrane orientation. We propose that, depending on the initial lipid bilayer geometry, DNA NPs can assemble into at least two different orientations: (i) as oligomers spanning planar membranes and (ii) in luminal, membrane-tethered orientation inside lipid nanostructures. While we found that both orientations coexist, substantial enrichment in lipid nanotubes suggests that the latter conformation may be either thermodynamically or kinetically favored. In the case of lipid nanotube networks, these membrane shaping functions probably synergize to not only stabilize lipid nanotubes from the inside of the lumen, but also to kink and branch these structures due to the high strains exerted upon transmembrane insertion.

The striking features of clustering and membrane remodeling achieved by a relatively simple DNA nanostructure underscores the pores' potential as molecular gatekeepers in membrane-based nanodevices. In particular, the ability to stabilize lipid nanotubes opens up new avenues to systematically control transport and communication in membrane-based networks found in several cell types[57, 58]. It may also be possible to mimic neurite-like connections by forming networks capable of transferring information or metabolites. DNA NP could furthermore help create drug-delivery or imaging vesicles that are stabilized by the DNA nanostructures, yet have designed permeability to release their cargo upon command. In these devices, the DNA structures would not only stabilize the lipid shape but also, via their pore function, act a selectivity filter for the transport. In conclusion, our discovery of the dual role of DNA NP is scientifically exciting and will impact their design of future biomimetic hybrid structures.

## Methods

**Design and assembly of DNA nanopores**. Information on the used modified and unmodified DNA sequences is provided in the Supplementary Information. An equimolar mixture of DNA duplexes (Supplementary Tables 1 and 2) dissolved in 15 mM Tris pH 8.0, 300 mM KCl was placed in a thermocycler at 95 °C for 10 min. Afterwards, the mixture was cooled with a constant rate of 0.25 °C min$^{-1}$ to 4 °C. The assembly of all used NPs was tested by agarose gel electrophoresis (1.2%) supplemented with 0.23% (w/v) sodium dodecyl sulfate. Five pmol of DNA was mixed with 5 μl SDS loading dye, before the gel was run at 60 V for 60 min at 8 °C. The NP integrity was analyzed after washing with deionized water for 15 min by appropriate fluorescence excitation for the used fluorescent dye conjugates and ethidium bromide staining.

**Surface modification for PSM assembly**. For microscopy experiments, PSM were assembled on standard glass cover slides (24 mm diameter) that were chemically modified with a PEG polymer brush and subsequently functionalized with hydrophobic anchors[30]. Briefly, the surfaces were cleaned by plasma treatment followed by silanization with (3-glycidyloxypropyl)trimethoxysilane at 75 °C for 1 h. After washing in dry acetone and drying under a nitrogen stream, the surfaces were reacted with pure molten diamino-PEG with a molecular mass of 2000 Da at 75 °C for 4 h. The surfaces were then reacted with palmitic acid solved in dimethyl sulfoxide mixed with an equimolar volume of pure N,N'-diisopropylcarbodiimide in presence of traces of N,N''-diisopropylethylamine for 30 min at room temperature, followed by extensive washing with chloroform.

TIRFS-RIf transducer slides were modified with a poly-L-lysine-graft-PEG copolymer carrying hexadecyl headgroups that spontaneously adsorbs to negatively charged surfaces and forms PSMs after subsequent vesicle binding[31]. The freshly cleaned transducers were briefly incubated with the polymer solution (1 mg ml$^{-1}$ solved in 50% (v/v) dimethylformamide in water) for 10 min, before TIRFS-Rif experiments were carried out.

**Lipid vesicle and polymer-supported membrane formation**. Very small unilamellar vesicles (VSUV) of DOPC or of DOPC supplemented with $^{OG488}$DHPE /1,1'-dioctadecyl-3,3,3',3'-tetramethylindodicarbocyanine (DiD) were formed by detergent extraction with heptakis(2,6-di-O-methyl)-β-cyclodextrin (β-CD)[59]. A mixture of 5 mM lipids and 20 mM Triton-X100 in HBS (20 mM Hepes pH 7.4, 300 mM NaCl) was incubated with a twofold excess of β-CD over the detergent for 5 min to form VSUVs and afterwards diluted to a lipid concentration of 250–500 μM. Chemically modified surfaces were incubated with these VSUVs for 30 min,

followed by removal of excess unbound vesicles. Bilayer formation was induced by incubating the surface with a 10% (w/v) solution of PEG8000 in HBS for 15 min. Extensive washing by strong pipetting with buffer removed the majority of excess lipid material from the bilayer.

**Binding kinetics by simultaneous TIRFS-RIf**. Simultaneous detection of TIRFS and mass-sensitive white light reflective interference (RIf) was performed in a home-built setup[60, 61]. The sample was formed by a 220 s long injection of 500 μM DOPC VSUVs, followed by rinsing of HBS buffer and an equally long injection of the respective DNA pore. During the dissociation phase, a constant flow of 10 μl s$^{-1}$ of HBS was applied. For each individual NP type and concentration used, the polymer surface was regenerated by two washing steps with 0.1% (v/v) of Triton-X 100. Extraction of pseudo-first-order kinetic constants was performed by separately fitting dissociation and association phases using BIAevaluation 3.1 software (GE Healthcare).

**Confocal imaging**. Confocal imaging was carried out with a confocal laser scanning microscope (Olympus FluoView 1000). OG488 was excited by the 488 nm laser line of a multiline argon laser (458/488/515 nm) and AF647 was excited by a 635 nm laser diode. Fluorescence was filtered by dichroic mirrors and spectral grating and collected between 505 and 600 nm for OG488, while AF647 emission was collected between 650 and 750 nm. To minimize fluorescence crosstalk, alternating excitation was used. Quantification of lipid and NP fluorescence in tubular and vesicular structures in respect to the plane membrane was performed after a global background-subtraction for a bleached area in each channel, followed by normalization in respect to the mean fluorescence of the planar membrane area.

**Single-molecule imaging by TIRFM**. TIRFM was carried out with an inverted Olympus IX71 microscope equipped with a quad-line total internal reflection illumination condenser (cellTIRF-4Line, Olympus) and a back-illuminated EMCCD camera (iXon Ultra 897, Andor Technologies). The sample was illuminated through a ×150 objective with a numerical aperture of 1.45 (UAPO TIRFM, Olympus). SRhoB was excited by a 561 nm diode-pumped solid-state laser (Cobolt Jive, 200 mW, Cobolt) with 2 mW output power at the objective, whereas FAM and AF647 were excited with 488 and 642 nm laser diodes (Omicron LuxX, Omicron Laserage Laserprodukte GmbH) with output powers of 5 and 4 mW at the objective, respectively. Fluorescence was imaged using a quadband emission filter (Brightline HC 446/523/600/677, Semrock) with an acquisition rate ranging from 33 to 91 fps.

Directly after formation of the PSM, the respective DNA NP was added at a concentration of 5 pM (single-molecule experiments) or 5 nM (confocal imaging) and incubated for 10 min. Unbound excess DNA NPs still in solution were then removed by extensive washing with buffer and an oxygen scavenging system composed of 0.5 mg ml$^{-1}$ glucose oxidase, 0.4 mg ml$^{-1}$ catalase and 4.5 mg ml$^{-1}$ glucose was added. Additionally, the redox-active photoprotectants ascorbic acid and methylviologen (each 1 mM) were supplemented[62]. Dual-color imaging of sulforhodamine B (SRhoB) and $^{AF647}$NP-3C was conducted after incubation with 2 μM of SRhoB for 30 s. After removal of excess SRhoB, imaging was commenced immediately. For all dual-color experiments, the camera chip was split into four different quadrants by a QuadView QV2 (Photometrics) spectral image splitter equipped with dichroic beam-splitters (565 and 640 nm longpass) and emission filters (BrightLine HC 600/37, BrightLine HC 685/40). Both fluorescence channels were manually aligned using fluorescent TetraSpeck microspheres (0.1 μm, Invitrogen) visible in all fluorescence channels.

**Single-molecule image analysis**. Localization of individual fluorescence emitters was performed by the multi-target tracking algorithm[63]. Before the localizations were subjected to frame by frame tracking, immobile molecules were filtered by a density-based spatial clustering of applications with noise (DBSCAN) algorithm[64]. Tracking was performed with the u-track algorithm developed by Jacqaman[65]. Diffusional properties of individual emitters were determined by fitting the MSD assuming free Brownian motion

$$\mathrm{MSD}(t) = d(2Dt + 2\varepsilon^2)$$

in one (nanotube, $d = 1$) or two dimensions (planar membrane, $d = 2$), where $D$ describes the diffusion coefficient and $\varepsilon$ comprises the localization precision. To extract the MSD, in each case the respective jump magnitudes were calculated from the individual trajectories and subsequently pooled. The MSD for increasing lag times was estimated from the probability density function of observed jump magnitudes as:

$$P(x,t) = \frac{\sqrt{2}}{\sqrt{\pi \mathrm{MSD}(t)}} e^{-\frac{x^2}{2\mathrm{MSD}(t)}}$$

within nanotubes where diffusion was restricted to 1 dimension, and

$$P(x,t) = \frac{x}{\mathrm{MSD}(t)} e^{-\frac{x^2}{2\mathrm{MSD}(t)}}$$

for 2D isotropic diffusion in the planar membrane. Diffusive sub-population could be identified by fitting a mixture-model of higher complexity if appropriate. The diffusion coefficient was determined by fitting the slope of MSD curve's first five frames (100 ms) for diffusion in nanotubes and 10–50 frames (200–1000 ms) for planar membrane diffusion. For dual-color co-localization, SRhoB/Cy3 and AF647 fluorescence channels were aligned based on a calibration matrix calculated from an image of TetraSpeck beads, which corrects for translations and rotations. Single-molecules within a distance of 107 nm (1 pixel) were considered co-localized. Counting of irreversible photobleaching steps was applied to resolve the number of DNA pores within individual diffraction-limited spots. Immobile DNA pore clusters were specifically identified in the time-averaged image (pixel-wise median of the first five frames) by their characteristic spatial constraint using the standard emitter localization scheme[63]. Extracted intensity-time trajectories were analyzed for photobleaching events using a generalized STaSI algorithm[37]. This algorithm models the raw data as noisy observations from a piece-wise constant signal. Irreversible photobleaching steps represent significant reductions in the time-averaged signal amplitude. Generally, the STaSI algorithm identifies such transition points in two phases, initially splitting the intensity-time trajectory into segments of constant amplitude, followed by clustering into an optimal number of statistically significant intensity levels. The initial state segmentation is calculated based on a two-sample Student's $t$-test with adaptive noise amplitude minimizing the probability for false partitioning. This partitioning process is performed until a limiting false-positive probability of 2% is reached. The final number of states was chosen based on the derived optimal minimum description length criterion for trajectories satisfying a minimum state separation of 70 photons. To allow robust molecule counting we filtered out fluctuations that can be ascribed to molecule-diffusion and fluorescence intermittency by applying a pre-defined cutoff-time of 50 frames as minimum fluctuation length. A second filter was put into place to limit the lag-time between successive bleaching steps (six frames). The so post-processed trajectories were then analyzed for their strictly decaying part to determine the final molecule count.

The dynamic nature of the formed immobile NP clusters was tested by spatially correlating the clusters in the same PSM area at two different time points by PICCS[39]. For this purpose, an area of the $^{AF647}$NP-3C loaded surface was imaged, followed by irreversible photobleaching of the pores in the area at 100% laser power for 1 min. After 15 min without illumination, the same area was imaged again and for both data sets, the immobile clusters were identified as described above. The significance of the results was tested using Monte Carlo simulations to estimate the amount of background correlation for the given cluster density and additionally by analyzing two uncorrelated regions of immobile NP-3C clusters with identical pore densities.

**Transmission electron microscopy**. Carbon-modified formvar-coated copper grids were plasma-activated under mild conditions and incubated with preformed VSUVs for 15 min. After removal of excess vesicles by washing with deionized water, the surface was incubated for 10 min with either 100 pM NP-3C or buffer. After continuous washing with deionized water without exposure to air, the surfaces were stained by 2% (w/v) uranyl acetate for 10 min and dried. TEM was conducted with a Zeiss 902 transmission electron microscope at 85 kV using a 2k CCD one-axis camera.

**Data availability**. The data that support the findings of this study are available from the authors on reasonable request.

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

## Acknowledgements

We thank Rainer Kurre for support with single-molecule fluorescence microscopy and Adrian Hodel for generating the images for the molecular models of DNA pores and membranes. This project was supported by the SFB 944 (P8 and Z) from the Deutsche

Forschungsgemeinschaft (to J.P.) and by UK EPSRC grant EP/N009282/1, and BBSRC grants BB/M025373/1 and BB/N017331/1 (to S.H.).

## Author contributions

O.B., J.R.B., S.H. and J.P. conceived the project and designed experiments. O.B. and J.R.B. performed experiments and evaluated the data. C.P.R. implemented evaluation algorithms and evaluated the data. O.E.P. performed TEM imaging. O.B., S.H. and J.P. wrote the manuscript with contributions of all authors.

## Additional information

**Competing interests:** The authors declare no competing financial interests.

