## [Peer Review File · Nature Communications]

Reviewers' comments:

Reviewer #1 (Remarks to the Author):

Howorka and Piehler present an interesting study of the interactions of DNA origami nanopores with the supported lipid bilayers formed on polymer brushes. The article reports several very interesting behaviors. First the authors show that the pores bind to the bilayer and insert in two phases, proceeding through an intermediate that represents the origami pore bound to the surface of the bilayer, which then converts to the inserted pore. The second very interesting behavior seems to be the stabilization of the DNA nanopore by curved bilayer and the formation of the lipid nanotubular structures with allegedly the DNA pores forming a scaffold in the center of the lipid tube.

The optical measurements presented in the work are well designed and perhaps represents state-of-the-art, the included supplementary movies are beautiful. However, some of the data (or more the inherent limitations of this optical probing approach), leave some open questions. The observation of the high density of the nanopores in the bilayer is remarkable, as the reported density of 14x14 nm membrane area per pore (which itself is 5x5nm!) is extremely high. That loading also means that almost 1/8 of the bilayer area is taken by the pores. What happens with the excess area of the lipid? Does the structure form "wrinkles", and if so, do those wrinkles lead to pore clustering? Evidence of the pore oligomerization reported in the manuscript likely points out to the possibility of the high-density pore array to form higher order structures or patterns. This system almost cries out for a high resolution imaging technique such, for example, cryogenic TEM or AFM (those techniques have a history of visualizing DNA origami structures, so suggesting them is not a stretch). NPs at the density of 14x14 nm/pore should be relatively easy to visualize with both techniques.

Perhaps the biggest revelation and the biggest weakness of this work comes from the observation of the lipid nanotube structures reinforced by the DNA NPs. The structure in which the NPs act as the anchors in the center of the tube would be remarkable, however the evidence for the existence of this structure is circumstantial at best, and at this point this structure remains merely a speculation. Again, a single TEM session could perhaps put those doubts to rest.

Even though this group has something very unique and potentially special with this system, I cannot support the publication in the present form. I however strongly urge this team to go back to the lab and try to obtain higher resolution structural information (especially a cryo-EM image of a lipid nanotube with DNA NPs inside). It would greatly enrich the manuscript, and I would be more than happy to give my strong support for this work once this higher resolution structural information is included.

Some additional specific comments on the text:

Abstract: a number of membrane remodeling proteins are known and some penetrate the membrane or have domains that penetrate the membrane, so the claim of unprecedented integration of membrane puncturing and remodeling functionality could be toned down.

Line 60. There are literature papers that discuss bilayer morphology changes after insertion of protein pores, for example, see Harroun, T. A.; Heller, W. T.; Weiss, T. M.; Yang, L.; Huang, H. W. *Biophys. J.* 1999, 76 (2), 937–945

Lines 114-115, also Fig 2D. Why do the authors see higher fluorescence signal from NP-1C than from NP-3C, especially if the 3C molecule has better anchoring?

Line 171-187 The whole paragraph is not quite clear. Is the claim that the NPs always assemble and disassemble in clusters with the same number of individual pores? If so, what is the mechanism of this behavior?

Line 209: Figure 5 already has membrane protrusions at time $t=0$, so it is not clear that those features are caused by the NPs.

Paragraph starting with line 211: What is the mechanism of pores being stable both in the bilayer and inside the nanotube. Usually biological proteins prefer one particular geometry.

Reviewer #2 (Remarks to the Author):

This manuscript describes experimental characterization of DNA nanopore (NP) interactions with lipid bilayer membranes. DNA NPs have previously been shown to spontaneously insert into lipid bilayers, providing passages for ions and other solutes. However, very little is known about the actual insertion mechanism, the spatial organization of the inserted DNA NPs within the bilayers and their effect on the lipid bilayer morphology. The manuscript reports new observations in all the three areas, elucidating a multi-step insertion process, aggregation of DNA NPs into clusters, and DNA NP-induced deformation of lipid membranes, including formation of lipid nanotubes. The study also presents a considerable advancement with regard to research methodology, adopting an arsenal of single molecule methods that have been previously used to study membrane proteins for the study DNA NPs.

Overall, this is, potentially, a very important study that can be of interest to researchers from the DNA nanotechnology, synthetic biology and membrane channel communities. The study, however, has several deficiencies that need to be addressed in the revised version.

The major problem is that it is not clear how all the phenomena described in the manuscript relate to another.

First, the authors use the incubation time to differentiate peripherally associated and fast moving NPs from those that insert into lipid bilayer and move much slower. However, how does this relate to cluster formation? The caption to Figure 3 states the incubation time of 1 hour, but the authors say (on line 129) that "longer incubation of more than 1h drastically decreased the mobility..." which we subsequently learn to be related to the formation of clusters. So how does one differentiate membrane insertion from cluster formation? Do they happen at the same time scale or does one follow the other?

Second, how does the process of lipid deformation characterized in Figure 5 relate to formation of lipid nanotubes, characterized in Figure 6? The authors change NP concentration, but are the two phenomena related? What is the conformation of NP when they deform lipid bilayer, specifically, are they inserted in the bilayer, do they form clusters or do they adopt yet another conformation?

Third, how can formation of lipid nanotubes, Figures 6, be reconciled with NPs insertion into the bilayer? The schematic shown in Figure 6 suggests that NPs are confined within the nanotube, but the sulforhodamine B assay (and its interpretation) requires NPs to span the lipid bilayer. Furthermore, NPs clustering is explained by a hydrophobic mismatch but that would require NPs to be inserted into the membrane.

Finally, how are the lipid nanotubes connected to the supported lipid bilayer? Are they not fully tubular or are they detached from the bilayer or are they protrusions that fell on the bilayer? Can this be resolved by super-resolution imaging?

A schematic chart placing different observation on the incubation time / NP concentration axes would be helpful. Ideally, the authors would support their observation with a free-energy analysis, although excessive theoretical description is clearly beyond the scope of this experimental study.

Minor comments:

It would be nice if authors could briefly state why polymer-supported membranes can represent

lipid membranes in vesicles and maybe mention some limitations of the model.

Caption to Figure 2: Spell out what PSM is.

Figure 3i: label the curves

Figure 3g caption: "Number of localized mobile and immobile ..." seems strange. Is "localized" used as a substitute for "observed"? Localized is synonymous to immobile, please revise.

Page 10, line 247: the authors describe a relative test where collisions for the one scenario are expected to be more frequent than for the other. But subsequently, the authors observe only one experimental outcome. The logic here is not clear.

Reviewer #3 (Remarks to the Author):

In the past years, DNA-membrane pores have been developed as a novel class of artificial membrane pores. Potential applications of such pores are found in biosensing or as membrane-penetrating agents. Even though membrane interactions and perforation have been shown in numerous experiments, the details of the interaction mechanism are not yet known.

In the present study, using single molecule biophysical methods, Birkholz et al. make an important contribution to the better mechanistic understanding of DNA membrane pores. The authors focus on small DNA pores formed by a short six-helix bundle hydrophobically functionalized with cholesterol at its circumference, but some of the insights of this study may be generalizable to other pores. In particular they observe fast binding of the nanopores to the membrane, insertion and clustering depending on the number of hydrophobic modifications, and also membrane reshaping. The claimed observation of lipid nanotubes and diffusion of pores within these tubes is quite stunning.

In a revised version of the manuscript, the authors should address the following points:

- the abbreviation PSM should be introduced properly
- maybe the authors could point out how important the PSM was for their observations – how would diffusion of the pores look like in a solid-supported lipid membrane?
- when the nanopores are inserted at their maximum density ($14 \times 14 \text{ nm}^2$), what happens to the lipids? Does the membrane expand, or are lipids expelled?
- binding of DNA nanopores and delayed insertion was previously observed in confocal microscopy experiments using other DNA pores and giant liposomes by Krishnan et al., Nature Comm. 7, 12787 (2016) (not cited in this manuscript), but not studied in greater detail as here.
- caption Figure 4b: "overlay of the STaSI trace shown in green" → red?
- the evidence for membrane reshaping is not overwhelming, but on the other hand quite conceivable as membrane deformation has been previously reported by others (e.g., Czogalla, A. et al. Angew. Chem. 127, 6601–6605 (2015)). In Fig. 5, the blobs are referred to as membrane protrusions – how do they look like and is there a way to estimate their size? Can they be elongated/deformed, e.g., in a flow?
- what is the size of the scale bar in Suppl. Figure 5? in this figure the authors refer to the "blobs" to as vesicular lipids rather than membrane protrusions (see previous comment). What is the difference?
- the observation of one-dimensional diffusion and its interpretation of diffusion of pores inside of tubes is amazing. How frequent did such nanotubes occur?
- are the diffusion properties inside of the nanotubes consistent with the expectation from the other measurements? The tubes seem to have a diffusion coefficient similar to that of membrane-inserted NP-3C, but the situation is different!

- others have generated lipid nanotubes by various other methods, which could be mentioned. It would be great to compare the properties with such nanotubes and potentially even generate and load them with pores (which probably is out of the scope of this article).

Reviewer #1 (Remarks to the Author):

Howorka and Piehler present an interesting study of the interactions of DNA origami nanopores with the supported lipid bilayers formed on polymer brushes. The article reports several very interesting behaviors. First the authors show that the pores bind to the bilayer and insert in two phases, proceeding through an intermediate that represents the origami pore bound to the surface of the bilayer, which then converts to the inserted pore. The second very interesting behavior seems to be the stabilization of the DNA nanopore by curved bilayer and the formation of the lipid nanotubular structures with allegedly the DNA pores forming a scaffold in the center of the lipid tube.

1.1: *The optical measurements presented in the work are well designed and perhaps represents state-of-the-art, the included supplementary movies are beautiful. However, some of the data (or more the inherent limitations of their optical probing approach), leave some open questions. This system almost cries out for a high resolution imaging technique such, for example, cryogenic TEM or AFM (those technique have a history of visualizing DNA origami structures, so suggesting them is not a stretch). NPs at the density of 14x14 nm/pore should be relatively easy to visualize with both techniques.*

Response: We thank the reviewer for this positive assessment of our work. We have carefully addressed the points raised by the reviewer. In particular, we focused on visualizing lipid nanotubes by AFM and TEM to complement the initial characterization by superresolution fluorescence microscopy (s. below).

Action: see below (1.3).

1.2: *The observation of the high density of the nanopores in the bilayer is remarkable, as the reported density of 14x14 nm membrane area per pore (which itself is 5x5nm!) is extremely high. That loading also mean that an almost 1/8 of the bilayer area is taken by the pores. What happens with the excess area of the lipid? Does the structure form “wrinkles”, and if so, do those wrinkles lead to pore clustering? Evidence of the pore oligomerization reported in the manuscript likely points out to possibility of the high-density pore array to form higher order structures or patterns.*

Response: Such very high densities of DNA NP at the membrane have solely been applied for the kinetic studies by reflectance interference spectroscopy as shown in Figure 2. The high concentrations were required to obtain reliable kinetic data (curvature) for the initial step of tethering to the PSM within a short injection period (< 5 min). Longer observation periods would have led other undesirable molecular processes (e.g. membrane-insertion of DNA NPs), which would have complicated the analysis of the initial membrane tethering.

We cannot rule out that DNA NPs at the very high density used for binding kinetics form additional membrane protrusions or higher-order structures. But we stress that the high density is not linked in any way to the DNA nanopore clusters observed in our manuscript. Rather, the clusters and the DNA-stabilized lipid nanotubes were obtained with 2500-fold lower densities of

~2 molecules/ μm^2 . These low densities are required by single molecule imaging and are also more in line with their use in biosensing and synthetic biology.

Action: In the revised manuscript, we have pointed out more prominently the different DNA NP densities used for different experiments.

1.3: Perhaps the biggest revelation and the biggest weakness of this work comes from the observation of the lipid nanotube structures reinforced by the DNA NPs. The structure in which the NPs act as the anchors in the center of the tube would be remarkable, however the evidence for the existence of this structure is circumstantial at best, and at this point this structure remains merely a speculation. Again, a single TEM session could perhaps put those doubts to rest. Even though this group has something very unique and potentially special with this system, I cannot support the publication in the present form. I however strongly urge this team to go back to the lab and try to obtain higher resolution structural information (especially a cryo-EM image of a lipid nanotube with DNA NPs inside). It would greatly enrich the manuscript, and I would be more than happy to give my strong support for this work once this higher resolution structural information is included.

Response: We agree that more detailed structural characterization of DNA-NP-induced lipid nanotubes would strengthen our study, and efforts have been made address this point (see below). But we disagree that the evidence of for the DNA-nanopore-lipid-tubes is circumstantial. Rather, our microscopic data characterize the nanotubes in terms of their dimensions and their adherence to the underlying supportive planar membrane. Furthermore, our study addresses the anchoring of DNA pores to the nanotubes and even the positioning of the DNA pores within the tube lumen.

Action: To gain more insights into the structural organization of lipid tubes, several experimental approaches were used. We first tried intensively to image lipid nanotubes by AFM, but the structures turned out to be too fragile and did not withstand the tapping motion of the AFM tip. We have included representative images as Reviewer Figure 1a,b to illustrate the rapid removal of lipid nanotubes during imaging.

Reviewer Figure 1. AFM images of lipid nanotubes. After acquiring the first image (a), lipid nanotubes are hardly detectable in the following image (b).

This observation is in line with additional FRAP experiments we performed in response to Reviewer 2 that show only few physical connections of lipid nanotubes with the underlying membrane (Supporting Figure 8 and Supplementary Movie 8, s. 2.3).

Transmission electron microscopy was used next to characterize DNA NP-induced lipid nanotubes. Strikingly, the lipid nanotubes could also be formed on membranes supported by carbon films that are required for imaging by negative stain TEM. Forming the lipid structure on EM grids was an important technical shift from the polymer-modified glass-slides used within the fluorescence microscopy studies. The TEM analysis clearly established the formation of DNA NPs-supported lipid nanotubes at ultrastructural level (new Figure 6f-h). The tubes' different organization as seen by fluorescence microscopy vs. TEM images can be explained by the switch from polymer-supported membranes to membranes directly bound onto EM grids.

1.4: Abstract: *a number of membrane remodeling proteins are known and some penetrate the membrane or have domains that penetrate the membrane, so the claim of unprecedented integration of membrane puncturing and remodeling functionality could be toned down.*

Response: We agree with the reviewer that integral membrane proteins can remodel bilayer geometry and therefore removed this statement from the abstract. Still we feel that unique features of bilayer remodeling by DNA NPs – in particular its ability to stabilize lipid nanotubes – result from its specific design that allows different geometries with respect to the bilayer.

Action: We have revised our discussion accordingly and also included further references including the one suggested by the reviewer (s. below) as well as a recent review about protein-based membrane curvature generation that was only mentioned in the discussion.

1.5: *Line 60. There are literature papers that discuss bilayer morphology changes after insertion of protein pores, for example, see Harroun, T. A.; Heller, W. T.; Weiss, T. M.; Yang, L.; Huang, H. W. Biophys. J. 1999, 76 (2), 937–945*

Response/Action: s. above (1.4).

1.6: *Lines 114-115, also Fig 2D: Why do the authors see higher fluorescence signal from NP-1C than from NP-3C, especially if the 3C molecule has better anchoring?*

Response: For the initial docking to the membrane (Figure 1c, step 1), the number of anchors apparently did not have a major impact in our experiments. For example, TIRFS-Rif experiments shown in Figure 2B indicate that the pore with one anchor has a slightly enhanced association rate compared to the three-anchored pores (cf. Supplementary Table 4). However, the referee is right to point out that Figure 2D suggests a bigger difference between pores with one and three anchors. We attribute this difference to the clustering of NP-3C within the polymer-supported membrane. Clusters comprising multiple DNA NP copies reduce the density of “localized molecules” in the identification process, as clusters are detected as individual molecules. Since the measurements were carried out 10 min after DNA NP incubation, substantial clustering of membrane-integral NP-3C is observed (Figure 3j-l). Furthermore, we cannot rule out that a small portion of pores already transiently oligomerize in solution and thus decrease the efficient concentration of free cholesterol moieties available for binding to the membrane. We minimized any DNA NP aggregation by assembling the pores directly before the experiments.

Action: In the revised manuscript, we have included these explanations for the different binding levels observed in Figure 2 c and d and furthermore added a stoichiometry analysis of the mobile DNA NPs as Figure 3j-l.

1.7: *Line 171-187: The whole paragraph is not quite clear. Is the claim that the NPs always assemble and disassemble in clusters with the same number of individual pores? If so, what is the mechanism of this behavior?*

Response: As we show in Figure 4a-d, the absolute number of NPs in the clusters is influenced by the overall NP density, which supports that the clustering is a random process. The important evidence from the experiments shown in Figure 4e-d is that the quasi-static, immobile clusters still dynamically exchange DNA NPs with the surrounding. This supports our model that DNA NP clustering is driven by hydrophobic mismatch, and that the cluster composition is dynamic.

Action: We have carefully revised this section and moved the explanation for DNA NP oligomerization in front of the section on FRAP experiments. Moreover, we have included the concept of liquid phase separation to describe the nature of DNA NP clusters.

1.8: *Line 209: Figure 5 already has membrane protrusions at time $t=0$, so it is not clear that those features are caused by the NPs.*

Response: Prior to the time series shown in Figure 5a, the surface was incubated with low nanomolar concentrations of ^{AF647}NP-3C for a brief time, followed by removal of excess pores. In other words, $t=0$ is not the time-point of adding the nanopores but the point when excess material was flushed away and fluorescence imaging was started.

Action: In order to make clear that the time-series started with DNA NPs already bound to the membrane, we have included images of the same surface before the NP-3C in the revised version of Figure 5a.

1.9: *Paragraph starting with line 211: What is the mechanism of pores being stable both in the bilayer and inside the nanotube. Usually biological proteins prefer one particular geometry.*

Response: We fully agree with the reviewer: the observation that DNA-NP can exist in two different orientations and thus also exert different functions. This is, in a nutshell, the important and surprising discovery of our study! We believe that the unusual dual role is caused by the unique interaction of the DNA NP's membrane-inserting hydrophobic anchors, which is different to conventional protein pores that have a continuous hydrophobic membrane-inserted surface.

Action: In our revised discussion, we have highlighted this key issue in more detail.

Reviewer #2 (Remarks to the Author):

This manuscript describes experimental characterization of DNA nanopore (NP) interactions with lipid bilayer membranes. DNA NPs have previously been shown to spontaneously insert into lipid bilayers, providing passages for ions and other solutes. However, very little is known about the actual insertion mechanism, the spatial organization of the inserted DNA NPs within the bilayers and their effect on the lipid bilayer morphology. The manuscript reports new observations in all the three areas, elucidating a multi-step insertion process, aggregation of DNA NPs into clusters, and DNA NP-induced deformation of lipid membranes, including formation of lipid nanotubes. The study also presents a considerable advancement with regard to research methodology, adopting an arsenal of single molecule methods that have been previously used to study membrane proteins for the study DNA NPs. Overall, this is, potentially, a very important study that can be of interest to researchers from the DNA nanotechnology,

synthetic biology and membrane channel communities. The study, however, has several deficiencies that need to be addressed in the revised version.

2.1: *The major problem is that it is not clear how all the phenomena described in the manuscript relate to another. First, the authors use the incubation time to differentiate peripherally associated and fast moving NPs from those that insert into lipid bilayer and move much slower. However, how does this relate to cluster formation? The caption to Figure 3 states the incubation time of 1 hour, but the authors say (on line 129) that “longer incubation of more than 1h drastically decreased the mobility...” which we subsequently learn to be related to the formation of clusters. So how does one differentiate membrane insertion from cluster formation? Do they happen at the same time scale or does one follow the other?*

Response: We agree with the reviewer that further clarification can help to better convey the unexpectedly complex interaction of DNA nanopores with membranes. In direct response to the reviewer’s comments, we find that DNA NP’s integration into membranes is rapidly followed by clustering, as inferred from the quantification of the NP-3C diffusion properties directly after incubation and after 1 h (Figure 3a, b), in conjunction with the intensity analysis of immobile molecules (Figure 4a-d). Any further side-by-side comparison is difficult given the different kinetics of the first-order process of membrane insertion and a second- or even multiple-order process of NP clustering.

Action: (i) We have included an overview in Figure 1 to visualize our findings about membrane interaction of DNA nanopores. (ii) We have carefully revised the results and discussion section.

2.2: *Second, how does the process of lipid deformation characterized in Figure 5 relate to formation of lipid nanotubes, characterized in Figure 6? The authors change NP concentration, but are the two phenomena related? What is the conformation of NP when they deform lipid bilayer, specifically, are they inserted in the bilayer, do they form clusters or do they adopt yet another conformation?*

Third, how can formation of lipid nanotubes, Figures 6, be reconciled with NPs insertion into the bilayer? The schematic shown in Figure 6 suggests that NPs are confined within the nanotube, but the sulforhodamine B assay (and its interpretation) requires NPs to span the lipid bilayer. Furthermore, NPs clustering is explained by a hydrophobic mismatch but that would require NPs to be inserted into the membrane.

Response: The processes of lipid budding (Figure 5) and nanotube formation (Figure 6) are related by excess lipid present in vesicular structures on the polymer supported membrane. Excess lipid and bilayer protrusions remain attached to the PSM after vesicles are fused on the polymer layer (cf. Roder *et al.*, Anal Chem 2011). Excess lipid was completely removed via washing prior to carrying out experiments on DNA NP insertion and clustering. However, we omitted such intense washing steps to test DNA NP binding to curved membranes. This not only increased binding of DNA NPs to protruding membranes, but also led to bilayer reshaping and the formation of lipid nanotube (Supplementary Movie 5 and 6). We therefore assume that

binding and reshaping curved membranes as well as lipid nanotube formation are related processes caused by DNA nanopores binding in a tethered, non-membrane spanning conformation. In the case of lipid nanotubes, our data furthermore demonstrate that the DNA NPs are localized within the lumen of the tube, which would perfectly match the hydrophobic properties of NP-3C. However, we cannot exclude that some cases DNA NPs are inserted into the membrane. This is supported by the observation that fluorophore sulforhodamine B is taken up into lipid nanotubes, likely via membrane-spanning pores, as pointed out by the reviewer. Consequently, our model is that two energetically favored states are possible for the triple-cholesterol-modified NP-3C: clusters of membrane-inserted pores as well as a membrane-adhering orientation which - in the context of highly curved membranes- allows anchoring of all three cholesterol moieties into the surrounding highly curved bilayer.

Action: In the revised manuscript, we have carved out more concisely our interpretation of the different geometries observed under different conditions. As pointed out in 2.1, we have included these interpretations already into our brief summary shown in Figure 1.

2.3: *Finally, how are the lipid nanotubes connected to the supported lipid bilayer? Are they not fully tubular or are they detached from the bilayer or are they protrusions that fell on the bilayer? Can this be resolved by super-resolution imaging?*

Response: We agree with the reviewer that the connection between lipid nanotubes and the underlying membrane is an important question to be addressed in the revised manuscript.

Action: We explored the connectivity of lipid nanotubes with the underlying PSM by FRAP of a fluorescent lipid probe. To this end, we completely bleached the fluorescence of lipid nanotubes, so that fluorescence recovery was only possible via connections with the planar membrane. These experiments suggest that nearly all lipid nanotubes are still connected to the underlying membrane, albeit only at a few sites along the tubular structures. In one example, we encountered a lipid nanotube that did not show fluorescence recovery suggesting that in rare cases fission of these structures is possible.

Action: We have included representative data as Supplementary Figure 8 and Supplementary Movie 8 in the revised manuscript.

2.4: *A schematic chart placing different observation on the incubation time / NP concentration axes would be helpful. Ideally, the authors would support their observation with a free-energy analysis, although excessive theoretical description is clearly beyond the scope of this experimental study.*

Response: We agree that modeling these structures and quantifying free energies is very important. Initial attempts to model membrane insertion of DNA NPs nicely support our concepts (Maingi *et al.*, *Nature Comm.* 7, 14784). But given the high complexity of the supramolecular structures as found by our single molecule imaging, a comprehensive computational study is

beyond the scope of our report. Moreover, we argue that the resulting DNA NP geometry is kinetically as well as thermodynamically controlled, which further complicates computational simulations.

Action: Within the cartoons shown in Figure 1d-f, we have included an overview, which of these structures is observed under which condition. We have also included the energetic considerations into the discussion.

2.5: *It would be nice if authors could briefly state why polymer-supported membranes can represent lipid membranes in vesicles and maybe mention some limitations of the model.*

Response: As already mentioned (above 2.2), PSM show membrane protrusions directly after vesicle fusion. Using these PSM as a model system was absolutely key to our discoveries as they not only allowed unraveling DNA NP integration and clustering at single molecule level, but also revealed membrane shaping activity upon binding to vesicular structures. The main bias introduced by the PSM architecture probably is that DNA NP clusters are more rapidly immobilized than in free-standing membranes due to interactions with the polymer cushion and its membrane anchors. In non-supported membranes that maintain mobility of clusters, we would expect that small clusters fuse into larger entities, similar as observed for phase-separating lipid membranes. For the purpose of unraveling the process of cluster formation however, immobilization of DNA NP clusters in PSM and thus stopping the reaction was even helpful.

Action: We have reflected on some unique features and drawbacks of PSMs in the introduction and the discussion of the revised manuscript (s. also 2.1). Moreover, we have performed additional single molecule tracking experiments with NP-3C binding to solid-supported membranes (new Supplementary Figure 5). Under these conditions, only negligible membrane insertion and clustering is observed, highlighting the key importance of the polymer cushion to study DNA NP.

2.6: *Caption to Figure 2: Spell out what PSM is.*

Response/Action: Polymer supported membrane is now spelled out in the introduction and the abbreviation PSM is introduced.

2.7: *Figure 3i: label the curves*

Response/Action: A legend was added into Figure 3i.

2.8: *Figure 3g caption: “Number of localized mobile and immobile ...” seems strange. Is “localized” used as a substitute for “observed”? Localized is synonymous to immobile, please revise.*

Response: Indeed we use the term “localized” synonymous to “detected”. The term “localized molecules” refers to the investigation by single molecule localization microscopy (SMLM) that is fundamentally based on the detection and localization of diffraction-limited signals. The attribute “mobile” or “immobile” is based on subsequent tracking analysis. Nanometer-sized clusters of multiple molecules, however, also yield diffraction-limited signals that are identified as individual “localized molecules”. We therefore prefer the technical term “localized molecule” in this context to avoid misunderstandings.

Action: We have included an explanation of the term “localized molecules” into the revised manuscript.

2.9: *Page 10, line 247: the authors describe a relative test where collisions for the one scenario are expected to be more frequent than for the other. But subsequently, the authors observe only one experimental outcome. The logic here is not clear.*

Response: The two scenarios refer to collisions of DNA NPs that are either diffusing along the nanotube (1D) or within the planar bilayer (2D). We observe collisions in 1D but not 2D. Collisions in 2D that yield a 180° change in direction is extremely unlikely considering the low density of DNA NPs on the surface and the molecules’ small diameter (~50-times smaller than the diffraction-limited signal for fluorescence imaging). Furthermore, collision in 2D is not observed because fluorophores photobleach prior to any molecular encounter. This is also the reason why we cannot monitor the clustering process (which requires collisions) at the single molecule level, but rather characterize cluster properties after formation. By contrast, our observation of molecular collisions within the movie strongly suggests that molecules diffuse in 1D within the linear lipid nanotubes, where their collision probability is substantially enhanced.

Action: We have included this explanation into the revised manuscript.

Reviewer #3 (Remarks to the Author):

In the past years, DNA-membrane pores have been developed as a novel class of artificial membrane pores. Potential applications of such pores are found in biosensing or as membrane-penetrating agents. Even though membrane interactions and perforation have been shown in numerous experiments, the details of the interaction mechanism are not yet known.

In the present study, using single molecule biophysical methods, Birkholz et al. make an important contribution to the better mechanistic understanding of DNA membrane pores. The authors focus on small DNA pores formed by a short six-helix bundle hydrophobically functionalized with cholesterol at its circumference, but some of the insights of this study may be generalizable to other pores. In particular they observe fast binding of the nanopores to the membrane, insertion and clustering depending on the number of hydrophobic modifications, and

also membrane reshaping. The claimed observation of lipid nanotubes and diffusion of pores within these tubes is quite stunning. In a revised version of the manuscript, the authors should address the following points:

3.1: *The abbreviation PSM should be introduced properly. Maybe the authors could point out how important the PSM was for their observations – how would diffusion of the pores look like in a solid-supported lipid membrane?*

Response: We only tested DNA NP integration into PSM, but not solid-supported membranes (SSM) because we anticipated that the strong negative charge of the glass surface in conjunction with the limited space between membrane and support would not allow the negatively charged DNA nanopores to integrate into the SSM membrane. Stimulated by the reviewers comment, we have now carried out the experiments with SSM and confirmed that DNA NP integration and clustering is obstructed in this membrane (s. also 2.5).

Action: We have introduced the abbreviation of PSM in the manuscript twice, and included additional experiments on DNA NP incorporation into SSM as Supplementary Figure 5.

3.2: *When the nanopores are inserted at their maximum density ($14 \times 14 \text{ nm}^2$), what happens to the lipids? Does the membrane expand, or are lipids expelled?*

Response: As already discussed in more detail above (1.2), we only used these very high densities for quantifying and comparing the membrane association kinetics of different DNA NPs. The binding curves indicate that even DNA-NP/PSM architectures remain stable at the high membrane loading. For the application in synthetic biology or bioensing, however, such high densities are not desired and therefore we focused on studying the spatiotemporal organization at relevant densities of a few DNA NP/ μm^2 .

Action: We have more stringently pointed out the DNA NP densities used in different experiments and the reasons for applying such densities.

3.3: *Binding of DNA nanopores and delayed insertion was previously observed in confocal microscopy experiments using other DNA pores and giant liposomes by Krishnan et al., Nature Comm. 7, 12787 (2016) (not cited in this manuscript), but not studied in greater detail as here.*

Response: We fully agree with the reviewer that the insertion process of other DNA NPs has been described before. The merit of our study however, is the detailed kinetic analysis of the two-step insertion and the discovery of much more complex and completely unexpected structural interplay between the DNA NPs and bilayer membranes. These striking features were not described in the study of Krishnan et al. (2016) as the design, dimension, and supramolecular organization of their “T-pores” is fundamentally different to our smaller pores.

Action: In the revised version, we have included the paper of Krishnan *et al.* (2016) within the introduction and the discussion, and we highlight the novelty of our work compared to previous studies.

3.4: *Caption Figure 4b: “overlay of the STaSI trace shown in green” → red?*

Response/action: Thank you for noticing this error – we have corrected the figure caption in the revised version.

3.5: *The evidence for membrane reshaping is not overwhelming, but on the other hand quite conceivable as membrane deformation has been previously reported by others (e.g., Czogalla, A. et al. Angew. Chem. 127, 6601–6605 (2015)). In Figure 5, the blobs are referred to as membrane protrusions – how do they look like and is there a way to estimate their size? Can they be elongated/deformed, e.g., in a flow?*

Response: We totally agree with the reviewer that other DNA architectures have been very successfully employed for shaping membrane and we highlight this work in our introduction. The striking and unique property of our DNA NP, however, is the ability to adopt different geometries that yield different functions, depending on the properties of the target membrane. Since we have observed very different types of protrusions induced by DNA NPs - as shown in Figure 5) - we have focused on the characterization of lipid nanotubes that appeared most unique and most reproducible.

Action: In the discussion of the revised version, we have more clearly highlighted the differences between our DNA NPs and previously published work on DNA-based membrane functionalization.

3.6: *What is the size of the scale bar in Suppl. Figure 5? in this figure the authors refer to the “blobs” to as vesicular lipids rather than membrane protrusions (see previous comment). What is the difference?*

Response: The use of the term “vesicular lipids” in this figure legend is indeed inconsistent with our terminology and was therefore changed.

Action: We have changed the “vesicular lipids” into “membrane protrusions” in the legend of this figure, which corresponds to Supplementary Figure 6 of the revised manuscript. We have also included the description of the scale bar’s dimensions.

3.7: *The observation of one-dimensional diffusion and its interpretation of diffusion of pores inside of tubes is amazing. How frequent did such nanotubes occur?*

Response: The occurrence of the lipid nanotubes is affected by both the amount of available lipid material and nanopores. As pointed out above (see 2.2), membrane protrusions and lipid nanotube formation required excess lipid at the PSM. The highest nanotube densities were reached, when the vesicles were premixed with equimolar ratio of the NP-3C prior to capturing to the substrate and subsequent fusion.

Action: We have included a representative image of lipid nanotubes generated under optimum conditions as Supplementary Figure 7 in the revised version of the manuscript.

3.8: *Are the diffusion properties inside of the nanotubes consistent with the expectation from the other measurements? The tubes seem to have a diffusion coefficient similar to that of membrane-inserted NP-3C, but the situation is different!*

Response: We explain the decreased diffusion constant of DNA NPs in lipid nanotube by the increased friction caused by all three cholesterol anchors being inserted into the membrane. Moreover, more frequent collisions with other DNA NPs inside the lipid nanotube may also contribute to a reduced diffusion constant.

Action: We have included some of these considerations into the revised version.

3.9: *Others have generated lipid nanotubes by various other methods, which could be mentioned. It would be great to compare the properties with such nanotubes and potentially even generate and load them with pores (which probably is out of the scope of this article).*

Response: We agree with the reviewer that it is important to refer to previous approaches to produce lipid nanotubes.

Action: In the discussion of the revised manuscript, we briefly point out the specific differences of DNA NP-induced lipid nanotubes compared to previous approaches.

REVIEWERS' COMMENTS:

Reviewer #1 (Remarks to the Author):

I applaud the authors for taking all of the reviewers' comments seriously and performing additional experiments to clarify some of the questions. I feel that the authors have addressed most of my comments, except the comment about the structure of the lipid tube reinforced with DNA NPs. The new negative-staining TEM images are striking. They really add a lot to the manuscript, and authors get strong credit for doing this (and other) follow-up studies. However, I still do not see how the Figure 6G and 6 E show the same structure. In fact, I do not see much evidence for the presence of DNA NPs inside the tube (I may see some hints of it, if I squint). Higher resolution TEMs of course would be very interesting to look at, but I do not know the limitations of the setup available to the authors. On the other hand, the kinks in the lipid tubes that we see in the TEM images (and there is no doubt that those kinks are a clear and a significant feature) are remarkable, and suggest that the DNA pore insertion alters the local geometry of the lipid bilayer and may lead to some of the effects seen in the work. My suggestion is to consider those questions and add a discussion of some alternative structures and how the TEM images fit with the rest of the data. After that discussion is added, I would be inclined to support publication. It is an interesting and thought-provoking work.

P.S. The PDF file uploaded to the system had serious formatting issues, the copy I downloaded had three instances of the Figure 6. Fortunately, a Word version did not have that issue.

Reviewer #2 (Remarks to the Author):

The authors have adequately addressed all comment of this reviewer.

Despite some technical problems with figure insertion (Figure 6 appears 12 times in the main text), the manuscript was a pleasure to read and can be accepted as is.

Reviewer #3 (Remarks to the Author):

The authors have responded to all of the referees' comments in a satisfactory manner and also added new data to further support their claims. This referee does not have any further requests and recommends publication in this form.

Reviewer #1 (Remarks to the Author):

I applaud the authors for taking all of the reviewers' comments seriously and performing additional experiments to clarify some of the questions. I feel that the authors have addressed most of my comments, except the comment about the structure of the lipid tube reinforced with DNA NPs. The new negative-staining TEM images are striking. They really add a lot to the manuscript, and authors get strong credit for doing this (and other) follow-up studies. However, I still do not see how the Figure 6G and 6 E show the same structure. In fact, I do not see much evidence for the presence of DNA NPs inside the tube (I may see some hints of it, if I squint). Higher resolution TEMs of course would be very interesting to look at, but I do not know the limitations of the setup available to the authors. On the other hand, the kinks in the lipid tubes that we see in the TEM images (and there is no doubt that those kinks are a clear and a significant feature) are remarkable, and suggest that the DNA pore insertion alters the local geometry of the lipid bilayer and may lead to some of the effects seen in the work. My suggestion is to consider those questions and add a discussion of some alternative structures and how the TEM images fit with the rest of the data. After that discussion is added, I would be inclined to support publication. It is an interesting and thought-provoking work.

Response: We thank the reviewer for appreciating our efforts. We agree that the TEM images themselves do not prove that DNA NPs are located inside lipid nanotubes, but rather confirm the formation of ultrathin lipid nanotubes. However, in conjunction with the single molecule diffusion and collision data, our evidence that DNA NPs can be located and diffuse inside lipid nanotubes is very strong. We also agree with the reviewer that the kinks (and branches) of lipid nanotubes is a characteristic feature of the lipid nanotube networks that we observe. We also agree that this feature can be explained by membrane-inserted DNA NPs and that this needs to be properly discussed within the manuscript. This feature actually nicely highlights how the combination of different membrane insertion geometries of DNA NPs is responsible for generating novel nanostructures.

Action: We have included these considerations into the revised manuscript (see blue text).